# The Effect of House Dust Sensitization on Skin Sebum and Moisture in Children with Allergic Respiratory Diseases

**DOI:** 10.3390/children10091483

**Published:** 2023-08-30

**Authors:** Uğur Altaş, Zeynep Meva Altaş, Nazlı Ercan, Mehmet Yaşar Özkars

**Affiliations:** 1Department of Pediatric Allergy and Immunology, Ümraniye Training and Research Hospital, University of Health Sciences, Ümraniye, 34764 Istanbul, Türkiye; myozkars@gmail.com; 2Ümraniye District Health Directorate, Ümraniye, 34764 Istanbul, Türkiye; zeynep.meva@hotmail.com; 3Department of Pediatric Allergy and Immunology, Gülhane Education and Research Hospital, University of Health Sciences, Etlik, 06010 Ankara, Türkiye; drnazliercan@gmail.com; 4Department of Pediatric Allergy and Immunology, Istinye University Faculty of Medicine, Zeytinburnu, 34010 Istanbul, Türkiye

**Keywords:** house dust mite, allergy, children, skin sebum, skin moisture

## Abstract

This study aimed to investigate the levels of skin moisture and sebum in children with a house dust allergy without skin symptoms. This was a case–control study involving children, aged 0–18 years, who were being followed up for an allergic airway disease in a pediatric allergy clinic. Age, gender, hemogram parameters, and IgE values were evaluated. The skin moisture and sebum percentages of the patients and control group were measured by a non-invasive bioimpedance method using a portable digital skin moisture and sebum measurement device on the cubital fossa. The median value of the skin moisture percentage in the house dust mite allergy-positive patient group was significantly lower than that in the house dust mite allergy-negative patients and the control group (*p* < 0.001). The house dust mite allergy-positive patient group had the lowest skin sebum content. However, there was no statistical significance among the groups in terms of skin sebum percentage (*p* = 0.102). In the study, children with a house dust allergy were found to have lower levels of skin moisture and sebum. The regular use of moisturizers for children with a house dust allergy should be kept in mind as an effective solution to protect the skin barrier and reduce skin symptoms.

## 1. Introduction

House dust mites are counted among the most prevalent allergens that can lead to respiratory diseases with allergic reactions [1,2]. Allergic reactions to house dust mites are quite common in children and may contribute to the development of allergic diseases, including asthma, allergic rhinitis, and atopic dermatitis [3,4]. House dust mite allergies are observed more frequently in the fall and spring months [5].

In the literature, there are many studies on house dust mite-induced allergic reactions in children [6,7,8]. In most of the studies conducted in our country and in different countries, the house dust mite has been reported as the most common aeroallergen in children [9,10,11]. In a large sample study conducted in China, house dust mites were reported as the most common allergen, with a rate of 29.7% in children [12]. In research carried out among asthmatic children in Türkiye, the frequency of house dust mite allergies in preschool and school-age children was reported to be 42.0% and 51.0%, respectively [13]. In a study conducted in Istanbul, the house dust mite was reported as the most common respiratory allergen in children [14].

Major allergens linked to allergic conditions, like asthma, allergic rhinitis (AR), and atopic dermatitis (AD), are primarily house dust mites, specifically Dermatophagoides farinae (Der f) and Dermatophagoides pteronyssinus (Der p) [15]. These mites contain potent allergens, proteases, and elements found in bacterial (lipopolysaccharide, LPS) and fungal (β-D-glucan) cell walls. These elements are recognized allergic triggers and robust immunomodulators. Der p, a significant house dust allergen, possesses robust protease activity that can disrupt the respiratory epithelium and intercellular connections within the skin [16]. Recent findings have unveiled the presence of mites within the gastrointestinal tract of healthy individuals, indicating their normal status within the human microbiome [17]. Their role in mending house dust-induced damage within the intestinal mucosa, resulting from chronic exposure to house dust allergens, potentially triggers regulatory mechanisms characterized by inflammatory cytokine production [18]. Furthermore, evidence suggests that house dust mites might contribute to the pathogenesis of intestinal destruction by compromising tight junctions between group 1 major allergens with cysteine protease activity and epithelial cells [19]. Apart from their significant impact on respiratory and gastrointestinal ailments, mites are recognized culprits in the development of atopic dermatitis, acne, and rosacea in human skin [20,21].

The skin acts as a barrier against microorganisms, allergens, and irritants from infiltrating the body. Therefore, the skin needs to maintain its ability to be a strong barrier [22]. The skin’s natural surface is characterized by an acidic, high-salt, dry, and aerobic environment, whereas follicle-sebaceous units lean towards being anaerobic and lipid-rich [23]. The body’s surface plays host to a complex array of flora, encompassing bacteria, fungi, and viruses. This symbiotic microbiome is crucial for upholding the skin’s barrier function by participating in key physiological processes [23,24]. The function of filaggrin (FLG) proteins is vital for preserving normal epidermal structure and skin barrier integrity. Filaggrin undergoes degradation to yield hygroscopic amino acids that form part of the skin’s natural moisturizing factors (NMFs) [25]. While the genetic, inflammatory, and drug-related factors underlying epidermal FLG deficiency are well established, limited information exists regarding environmental stressors, such as house dust, that could potentially reduce FLG and NMF levels [26].

Exposure of the skin to barrier-disrupting substances, or some genetically deficient properties of the barrier molecules, cause the opportunistic pathogens to colonize on the skin surface, and epithelial inflammation can begin [27]. Factors such as air pollution, climatic changes, microplastics, smoking, and dietary habits may damage the epithelial barrier of the skin and mucosal surfaces. This impairment has been linked to a heightened occurrence and intensity of conditions marked by allergy and inflammation, including atopic dermatitis, food allergies, allergic rhinitis, and asthma [28,29].

Moisture and sebum balance of the skin is necessary for barrier properties. When this balance is disrupted, symptoms such as dryness occur on the skin [30,31]. Dry skin is a common symptom of atopic dermatitis, which is one of the allergic diseases [32]. There is an impaired skin barrier in atopic dermatitis [30,33]. The pathophysiologic mechanisms of allergic diseases are similar to each other [34]. The allergic march is the succession of atopic dermatitis, asthma, and allergic rhinitis [35,36]. Since allergic diseases may have similar clinical courses and pathophysiologic mechanisms, disruption of the skin barrier may be observed in allergic respiratory diseases even in the absence of accompanying dermal findings.

Skin health may be negatively affected in the presence of a house dust allergy [37,38]. A lack of moisture in the skin may lead to skin dryness and itching [39]. Therefore, the evaluation of skin moisture and sebum in children with a house dust mite allergy, even in the absence of skin symptoms, is important in terms of protecting and improving skin health. This study aimed to investigate the levels of skin moisture and sebum in children with a house dust allergy without skin symptoms. It also aimed to compare skin sebum and moisture levels with control groups consisting of children without a house dust mite allergy.

## 2. Materials and Methods

### 2.1. Study Design, Type and Sample

This was a case–control study involving children aged 0–18 years who were being followed up for an allergic airway disease in the pediatric allergy clinic of a tertiary health care institution. Patients were divided into two groups according to the presence of sensitization to house dust (Der f and/or Der p), which was determined through a skin prick test (SPT)/specific Ig E (Sp IgE). Group 1 patients (n = 114) were those with sensitization to house dust, while group 2 patients (n = 118) were children with allergic respiratory diseases without sensitization to house dust. The control group consisted of 110 age- and gender-matched healthy children who were admitted to the pediatric clinic of our hospital for minor surgical procedures, check-ups, and vaccinations. The children in the control group did not have any doctor-diagnosed allergic diseases. During the data collection period (May–June 2023), all participants who met the inclusion criteria of the study and gave consent were included.

Exclusion criteria were as follows: filaggrin mutations, xerosis, history of atopic dermatitis/present atopic dermatitis, inflammatory, genetic, and/or infectious diseases, (e.g., dermatitis herpetiformis, molluscum contagiosum, tinea, mycosis fungoides, psoriasis, urticaria and scabiasis, rheumatoid arthritis, and inflammatory bowel disease), use of special skin care, diet, multivitamins, antihistamines, or montelukast, or intense exposure to ultraviolet radiation within the last 4 weeks.

### 2.2. Evaluations

Sociodemographic characteristics (age, gender, etc.), hemogram parameters (leukocyte, eosinophil, lymphocyte, thrombocyte counts), and total IgE values were evaluated. Skin moisture and sebum levels of the patients and control group were measured using a portable digital skin moisture and sebum measurement device on the cubital fossa.

### 2.3. Measurement of Skin Sebum and Moisture

Prior to measurement, children rested for nearly 10–20 min to reduce the effect of physical activity on skin moisture and sebum levels.

Patients were instructed to abstain from applying moisturizing cream, sunscreen, other creams, baby oils, or using harsh exfoliants on the skin area at least one day prior to the skin moisture and sebum measurements. On the day of measurement, patients were instructed to cleanse the designated skin area with plain water at home. The measurements took place in an environment maintained at around 20 °C with the air humidity ranging from 40% to 60%. After each measurement, the device was cleaned with a soft cloth containing alcohol.

A portable pen-shaped Digital Skin Moisture Sebum Analyzer with LCD Display (Reyoung-Beauty, Guangdong, China) was used to measure the sebum and moisture levels of the skin. This device obtains measurements using the bioimpedance method, which is a non-invasive method. Bioimpedance, also referred to as biological impedance, delineates the capacity of biological tissue to hinder the flow of electric currents. This phenomenon reflects the passive electrical characteristics inherent in biological substances [40]. This device was produced commercially for the measurement of the moisture and sebum levels on the skin. Although this device has not been validated and is not used during diagnostic tests on patients with allergic diseases, the device performs measurement using the bioimpedance method, which is a non-invasive and inexpensive method [41], and the use of similar devices will be feasible in clinical practice. Previous studies in the literature have evaluated skin moisture and sebum content using this device [30,42].

Measurements were performed on the bare skin for a few seconds by placing the probe of the device on the antecubital fossa of the non-dominant upper limb. The device gives the results for sebum and moisture levels of the skin as percentages. The measurable moisture range is between 0% and 99.9%. The measurable sebum range is between 16.0% and 63.0%. The device is lightweight, portable, and easy to use. The dimensions of the device are 128 × 26 × 34 mm^3^.

### 2.4. Statistical Analysis

The data analysis and recording were conducted using the IBM SPSS 25.0 software program designed for Social Sciences for Windows. Descriptive statistics are presented as median, minimum, and maximum values, along with counts (n) and percentages (%). Categorized data were compared using the Chi-Square test. The normal distribution of continuous variables was assessed through both visual methods (histograms and probability plots) and statistical tests (Kolmogorov–Smirnov/Shapiro–Wilk). In cases where normal distribution was not met, the Mann–Whitney U test was employed to compare continuous variables, while the Spearman correlation analysis was chosen for the comparison of non-normally distributed continuous variables. ROC (Receiver Operating Characteristics) curve analysis was employed to assess the predictive potential of skin moisture (%), in relation to test positivity for house dust mites (confirmed via skin prick test and/or specific IgE). Sensitivity and specificity were reported upon identification of a significant cutoff point. Statistical significance was recognized at a *p* value below 0.05.

### 2.5. Ethics

Ethical approval was obtained from the Health Sciences University Ümraniye Training and Research Hospital Ethics Committee (Date: 11 May 2023, decision number: 141). Patients and parents were informed before their participation in the study, and informed consent for their participation in the study was obtained.

## 3. Results

The number of pediatric patients with allergic rhinitis and asthma who also had a house dust mite allergy was 114. The number of children with allergic rhinitis and asthma who had a negative house dust mite allergy test was 118. The healthy control group without any chronic disease or allergy diagnosis consisted of 110 children. The gender distribution was similar between the groups (*p* = 0.344). The median age of the house dust mite allergy-positive group was 7.0 years (1.0–18.0), while the median age of the house dust mite allergy-negative group and the healthy control group was 6.0 years (0–17.0). The one-year median age difference between the house dust mite allergy-positive and negative patient groups was statistically significant, whereas it was not significant in the healthy group (Table 1).

The percentages of skin moisture and skin sebum, and laboratory parameters of the patients and control group were evaluated. The median values of the skin moisture percentage in the house dust mite allergy-positive and negative patient group were 33.5% (10.0–50.0) and 37.0% (11.0–58.0), respectively. The skin moisture percentage was 38.0% (19.0–69.0) in the healthy control group. The median value of the skin moisture percentage in the house dust mite allergy-positive patient group was statistically significantly lower than that in the house dust mite allergy-negative patients and the control group (*p* < 0.001. Patients with a negative allergy test also had a lower median value of skin moisture percentage compared to the control group, but statistical significance was not observed (*p* = 0.120) (Table 2).

The house dust mite allergy-positive patient group had the lowest skin sebum content. The patient group with negative allergy tests also had a lower skin sebum percentage compared to the control group. However, there was no statistical significance between the groups in terms of the skin sebum percentage (*p* = 0.102) (Table 2).

Within the scope of the study, the WBC (white blood cell), neutrophil, eosinophil, lymphocyte, platelet, and total IgE values of the children were compared. No statistically significant difference was found between the groups in terms of WBC, neutrophil, lymphocyte, and platelet values (*p* > 0.05). However, the eosinophil and total IgE median values of the house dust mite allergy-positive patient group were statistically significantly higher than those of the allergy-negative patient group (*p* < 0.001) (Table 2).

The relationships between the skin moisture percentage, skin sebum percentage, eosinophil value, and IgE values of the house dust mite allergy-positive patient group were evaluated. A statistically significant negative correlation was observed between the skin sebum and skin moisture percentage (r = −0.304, *p* < 0.001). There was also a statistically significant negative correlation between the skin moisture percentage and total IgE values (r = −0.232, *p* = 0.001). There was no significant correlation between the skin sebum percentage and total IgE values (r = 0.034, *p* = 0.632). No statistically significant correlation was found between the eosinophil value and the skin moisture and sebum percentages (*p* > 0.05) (Table 3).

An ROC analysis was performed to evaluate the predictive value of the skin moisture percentage for the presence of a house dust mite allergy. According to the ROC analysis, taking 35.5% as the cut-off point for percent skin moisture, the sensitivity and specificity were 78.9% and 57.9%, respectively. The area under the curve was 73.0% (95.0% CI: 66.7–77.4%) (*p* < 0.001) (Figure 1).

## 4. Discussion

Disorders in the barrier function of the skin may be observed in allergic diseases [43]. Monitoring skin moisture and sebum levels and maintaining moisture and sebum balance in the skin are important in the control of allergic diseases. Although there are many studies evaluating skin moisture and sebum levels in patients with atopic dermatitis, studies evaluating skin moisture and sebum levels in patients with asthma and allergic rhinitis, which are other allergic diseases, are very limited. In this study, we evaluated skin moisture and sebum levels in asthma and allergic rhinitis patients with and without a house dust mite allergy, and in a control group.

In our study, the median value of the skin moisture percentage in the house dust mite allergy-positive patient group was significantly lower than that in the allergy-negative patient group and the control group. The percentage of skin moisture in the allergy-negative patient group was also lower than in the control group, although statistical significance was not observed. In other words, patients with allergic rhinitis and asthma, with or without a house dust mite allergy, had lower skin moisture levels than the healthy group. In patients with a house dust mite allergy, the lower skin moisture percentage was more pronounced. In our study, although statistical significance was not observed in the skin sebum percentage measurements, the lowest value was observed in the house dust mite allergy-positive patient group. In a study conducted in children diagnosed with atopic dermatitis in our country, skin moisture and skin sebum ratios of children with atopic dermatitis were found to be lower than those of healthy children [30]. Similarly, skin sebum was reported to be lower in patients with atopic dermatitis in another study [9]. Since the pathophysiologic mechanisms of allergic diseases are similar to each other, although there is no skin involvement in asthma and allergic rhinitis patients with house dust mite allergies, disruption in the skin barrier may also be observed in these patients [34]. This deterioration may be due to a decrease in the sebum and moisture content of the skin. Further studies are needed to more clearly explain the pathophysiologic mechanism of the decrease in moisture and sebum content in the skin of asthma and allergic rhinitis patients with a house dust mite allergy and without skin involvement. Another fact is that atopic dermatitis may be under-diagnosed, especially in mild cases, and this could explain our results. Our study’s results highlight that skin moisturizers should be recommended as a preventive measure in children with allergic diseases, regardless of skin involvement.

In our study, a statistically significant negative correlation was observed between skin sebum and skin moisture. The sebum content of the skin plays an important role in maintaining skin moisture and skin integrity [10]. To maintain the barrier function of the skin, the sebum and moisture content of the skin should be in a certain balance [30]. The negative correlation between skin moisture and skin sebum measurements is an expected situation in terms of ensuring moisture balance. The treatment of patients in allergy clinics with skin barrier-protective therapies, sebum balance-protective products, and skin moisturizers will help to support the protection of the skin’s barrier function and provide the skin with moisture balance. In our study, a statistically significant negative correlation was observed between skin moisture and total IgE levels. The increase in total IgE levels may be associated with the role of allergic mechanisms in patients. The fact that higher levels of IgE are observed in asthma patients with more severe clinical manifestations in the literature suggests that IgE levels may be related to the clinical severity of the disease [32]. In our study, the fact that those with higher IgE levels had lower skin moisture levels supports the hypothesis that the underlying allergic pathophysiologic mechanisms in allergic diseases reduce skin moisture.

In our study, an ROC analysis was performed to evaluate the predictive value of the skin moisture percentage for patients who have a house dust mite allergy. When 35.5% was taken as the cut-off point for skin moisture percentage, the sensitivity and specificity were found to be 78.9% and 57.9%, respectively. According to our results, 78.9% of patients with a house dust mite allergy can be identified by measuring skin moisture. This rate is approximately four out of five patients and is a very high rate. Establishing a cut-off point value for the percentage of skin moisture to predict a house dust mite allergy may be a good alternative for patients in whom the presence of a house dust allergy cannot be assessed by skin testing or specific IgE levels. The measurement of skin moisture and sebum by practical, easily applicable, and accessible, non-invasive methods will greatly facilitate the management of allergic diseases in clinical practice.

### Limitations and Strengths

There are studies on the relationship between allergic diseases and an impaired skin barrier. However, as we know, there are no studies examining the percentage of skin moisture and sebum in children with a house dust mite allergy who do not show skin manifestations and do not have any skin involvement. In this regard, our findings make an important contribution to the literature. On the other hand, in our study, the skin moisture and sebum measurements were performed from a single skin area, which may have limited the interpretation of the study’s results. Although the device we used is commercially available for skin sebum and moisture measurements and has been used in epidemiologic studies, the lack of a validation study may limit the interpretation of the results. Another limitation is that two different examiners analyzed skin moisture and sebum, and there might be an interobserver variability. Obviously, the children from the normal cohort could induce statistical errors because their parents paid little attention to preparing the skin area for testing sebum and moisture when compared to the patient group. Another limitation is that, since children may have allergies against multiple allergens, such as food allergens, which is commonly seen among children [44], we could not evaluate the effects of multiple allergies on the skin sebum and moisture contents. Further studies are needed to determine the effect of allergic diseases on skin moisture and sebum content more specifically. Validation studies using non-invasive devices measuring skin sebum and moisture levels in children with different allergic diagnoses can be conducted.

## 5. Conclusions

In this study, children with a house dust allergy were found to have a low skin moisture and sebum content when compared to healthy children. This may indicate that children with a house dust allergy have a weak skin barrier, resulting in increased water loss from the skin and thus, dry skin and itching may be observed in these children. In addition, a weak skin barrier can reduce the skin’s defenses against the substances that cause the house dust allergy and thus worsen the existing skin symptoms. For these reasons, the regular use of moisturizers for children with a house dust mite allergy should be kept in mind as an effective solution to protect the skin barrier and reduce skin symptoms.

## Figures and Tables

**Figure 1 children-10-01483-f001:**
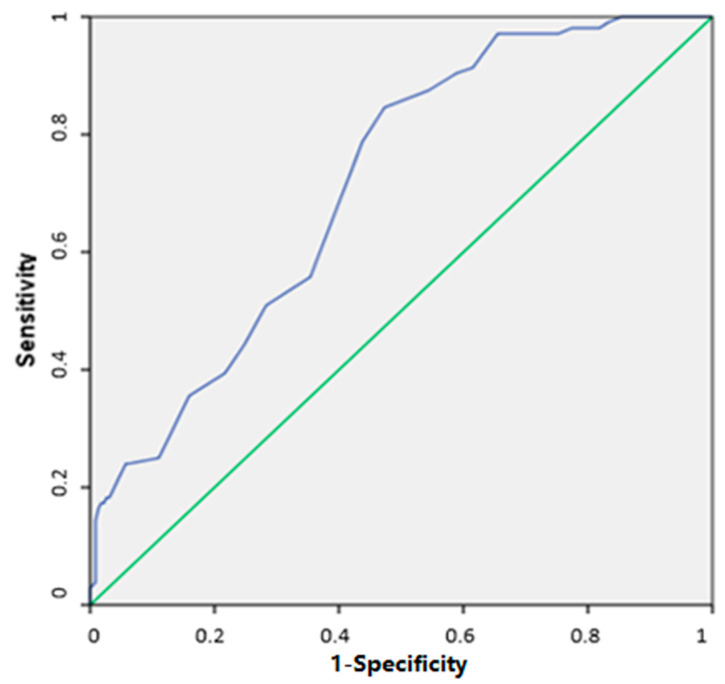
Skin moisture percentage predicts the presence of house dust mite allergy.

**Table 1 children-10-01483-t001:** Age and gender of the patients and control group.

	Group 1 (HDM Allergy)	Group 2 (No HDM Allergy)	Control Group	*p* Value
Age (years), median (min-max)	7.0 (1.0–18.0)	6.0 (0–17.0)	6.0 (0–17.0)	0.041
Gender, n (%)				
Female Male	53 (46.5) 61 (53.5)	54 (45.8) 64 (54.2)	60 (54.5) 50 (45.5)	0.344

HDM: House dust mite.

**Table 2 children-10-01483-t002:** Skin moisture, sebum, and laboratory values of the patient and control groups.

	Group 1 (HDM Allergy)	Group 2 (No HDM Allergy)	Control Group	*p* Value
	Median (Min–Max)	Median (Min–Max)	Median (Min–Max)	
Skin moisture (%)	33.5 (10.0–50.0)	37.0 (11.0–58.0)	38.0 (19.0–69.0)	<0.001
Skin sebum (%)	24.5 (16.0–55.0)	25.0 (16.0–49.0)	28.0 (16.0–51.0)	0.102
WBCs (10^3^ mm^3^)	8500.0 (4000.0–19,000.0)	8145.0 (3050.0–21,580.0)	-	0.172
Neutrophils (10^3^/uL)	4000.0 (1700.0–38,080.0)	3850.0 (300.0–10,370.0)	-	0.156
Eosinophils (10^3^/uL)	380.0 (40.0–2260.0)	220.0 (0.0–4160.0)	-	<0.001
Eosinophils (%)	4.4 (0.3–113.3)	2.8 (0.0–21.0)	-	<0.001
Lymphocytes (10^3^/uL)	3100.0 (1400.0–6500.0)	3210.0 (1102.0–7870.0)	-	0.404
Platelets (10^3^ mm^3^)	355,000.0 (150,000.0–659,000.0)	313,500.0 (190,000.0–734,000.0)	-	0.092
Total IgE (IU/mL)	304.0 (4.0–3840.0)	41.0 (1.0–3633.0)	-	<0.001

WBC: white blood cell, HDM: house dust mite.

**Table 3 children-10-01483-t003:** Correlation of skin moisture, sebum, eosinophil, and IgE values of the patient group.

		Skin Mositure	Skin Sebum
Skin Moisture	Correlation coefficient (r)	-	−0.304
	*p* value	-	<0.001
Skin Sebum	Correlation coefficient (r)	−0.304	-
	*p* value	<0.001	-
Eosinophils (10^3^/uL)	Correlation coefficient (r)	−0.084	−0.019
	*p* value	0.230	0.783
Eosinophils (%)	Correlation coefficient (r)	−0.089	−0.029
	*p* value	0.201	0.683
Total IgE (IU/mL)	Correlation coefficient (r)	−0.232	0.034
	*p* value	0.001	0.632

## Data Availability

Not applicable.

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
