# Peer review of "The Effect of House Dust Sensitization on Skin Sebum and Moisture in Children with Allergic Respiratory Diseases"

_children, 2023, doi:10.3390/children10091483_

Round 1
Reviewer 1 Report
The presented study demonstrate that patients with allergic rhinitis and asthma are , with or without house dust sensitivity (there is a differance between skin sensitivity as demostrated by SPT or specesific IGE in serum and a 'true allergy' meaning- clinical symtoms related to the suspected allergen exposure), have lower skin moisture compared with healthy controls.
I have 2 notes:
1- atopic dermatitis is under-diagnosed especiialy in mild cases and this could expalin the results
2- it should be discuss in the study whether according to the results it is recommended to treat patients with no skin complaints..
Author Response
Point 1: Atopic dermatitis is under-diagnosed especially in mild cases and this could explain the results.
Answer 1: Dear Reviewer, thank you for your valuable comments and suggestions. We have included this sentence at the bottom of the second paragraph in the discussion section.
Point 2: It should be discuss in the study whether according to the results it is recommended to treat patients with no skin complaints.
Answer 2: Thank you for your suggestion. We have included this sentence at the bottom of the second paragraph in the discussion section.
Reviewer 2 Report
In this study, Altas et al. decided to investigate the levels of skin moisture and sebum percentages in children with house dust allergy without skin symptoms. It was also aimed to compare skin sebum and moisture levels with control groups without house dust mite allergy.
The topic of the article is quite interesting, however my biggest concern is the method, which was used during the study.
According to my knowledge, this method was not validated and is not recommended during diagnostics of patients with AD.
In addition, the conclusions drawn in a study conducted using this research method are too far-reaching.
Author Response
Point 1: According to my knowledge, this method was not validated and is not recommended during diagnostics of patients with AD.
Answer 1: Dear Reviewer, thank you for your valuable comments and suggestions. We have included this information in the third paragraph of the “2.3. Measurement of skin sebum and moisture” section.
Point 2: In addition, the conclusions drawn in a study conducted using this research method are too far-reaching.
Answer 2: Thank you for your comment. We have mentioned this in the Limitations part of the manuscript with the sentences “Further studies are needed to determine the effect of allergic diseases on skin moisture and sebum content more specifically. Validation studies of non-invasive devices measuring skin sebum and moisture in children with different allergic diagnoses can be conducted.”
Reviewer 3 Report
Dear Authors,
I congratulate you on this complex study regarding allergies in children.
However, there are some aspects that require your attention.
You mention a portable digital skin moisture and sebum measurement device - please insert the commercial and technical description of the device.
In the discussion section you need to expand on other possible allergies of the children. Because rarely children have allergies only to one allergen. Reference this to the work by Berghi ON, Vrinceanu D, Cergan R, Dumitru M, Costache A. Solanum melongena allergy (A comprehensive review). Exp Ther Med. 2021 Oct;22(4):1061. doi: 10.3892/etm.2021.10495. Epub 2021 Jul 27. PMID: 34434275; PMCID: PMC8353643.
Expand the limitations section of the manuscript. For example the examiner was different when analizing skin moisture and sebum and there might be an interobserver variability. Obviously the children from the normal cohort could induce statistical errors because parents paid little attention to preparing the skin area for testing sebum and moisture.
I look forward to receiving the improved form of your manuscript.
Author Response
Point 1: You mention a portable digital skin moisture and sebum measurement device - please insert the commercial and technical description of the device.
Answer 1: Dear Reviewer, thank you for your valuable comments and suggestions. We have included this information in the third and fourth paragraph of the “2.3. Measurement of skin sebum and moisture” section.
Point 2: In the discussion section you need to expand on other possible allergies of the children. Because rarely children have allergies only to one allergen. Reference this to the work by Berghi ON, Vrinceanu D, Cergan R, Dumitru M, Costache A. Solanum melongena allergy (A comprehensive review). Exp Ther Med. 2021 Oct;22(4):1061. doi: 10.3892/etm.2021.10495. Epub 2021 Jul 27. PMID: 34434275; PMCID: PMC8353643.
Answer 2: Thank you for your comment. We have mentioned this in the Limitations part of the manuscript with the sentences “Another limitation is that since children may have allergies against multiple allergens such as food allergy which is commonly seen among children [44], we could not evaluate the effects of multiple allergies on the skin sebum and moisture contents.”
Point 3: Expand the limitations section of the manuscript. For example the examiner was different when analizing skin moisture and sebum and there might be an interobserver variability. Obviously the children from the normal cohort could induce statistical errors because parents paid little attention to preparing the skin area for testing sebum and moisture.
Answer 3: Thank you for your comment. We have mentioned these conditions in the Limitations part of the revised manuscript.
Round 2
Reviewer 2 Report
Thank you for the revised manuscript.
Unfortunately, my doubts about the methodology used in the study have not been dispelled.